# From Speaker to Dubber: Movie Dubbing with Prosody and Duration Consistency Learning

## ABSTRACT

Movie Dubbing aims to convert scripts into speeches that align with the given movie clip in both temporal and emotional aspects while preserving the vocal timbre of one brief reference audio. The wide variations in emotion, pace, and environment that dubbed speech must exhibit to achieve real alignment make dubbing a complex task. Considering the limited scale of the movie dubbing datasets (due to copyright) and the interference from background noise, directly learning from movie dubbing datasets limits the pronunciation quality of learned models. To address this problem, we propose a two-stage dubbing method that allows the model to first learn pronunciation knowledge before practicing it in movie dubbing. In the first stage, we introduce a multi-task approach to pre-train a phoneme encoder on a large-scale text-speech corpus for learning clear and natural phoneme pronunciations. For the second stage, we devise a prosody consistency learning module to bridge the emotional expression with the phoneme-level dubbing prosody attributes (pitch and energy). Finally, we design a duration consistency reasoning module to align the dubbing duration with the lip movement. Extensive experiments demonstrate that our method outperforms several state-of-the-art methods on two primary benchmarks. The source code and model checkpoints will be released to the public. The demos are available at https://speaker2dubber.github.io/.

## CCS CONCEPTS

• **Computing methodologies → Phonology / morphology**; *Computer vision.*

## KEYWORDS

Movie dubbing, visual voice cloning, two-stage framework

**ACM Reference Format:**
Anonymous Authors. 2024. From Speaker to Dubber: Movie Dubbing with Prosody and Duration Consistency Learning. In *Proceedings of the 32nd ACM International Conference on Multimedia (MM'24), October 28-November 1, 2024, Melbourne, Australia.* ACM, New York, NY, USA, 10 pages. https://doi.org/10.1145/nnnnnnn.nnnnnnn

## 1 INTRODUCTION

Movie Dubbing, also known as Visual Voice Cloning (V2C) [4], aims to convert scripts into speeches utilizing a specified voice guided by a short reference audio while maintaining lip synchronization

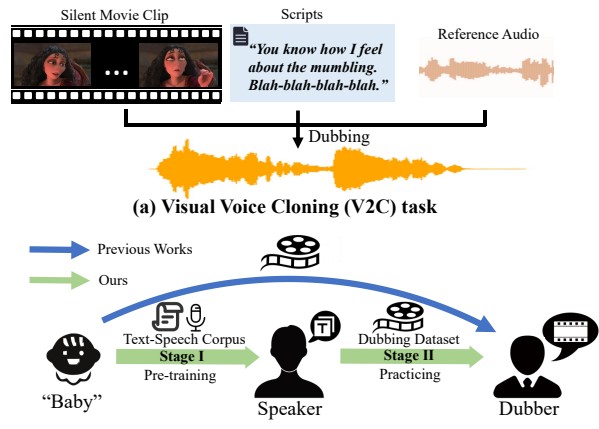

**Figure 1: (a) Illustration of the V2C task. (b) To overcome the limitations of movie dubbing datasets in scale and enable the model to generate dubbing with accurate and clear pronunciation, we propose a novel two-stage dubbing method. Unlike previous single-stage methods, it involves the pre-training stage for pronunciation knowledge and the practicing stage for dubbing abilities.**

and reflecting the character's emotions in the given movie clip (as shown in Figure 1 (a)). It requires the ability to bridge the visual, natural language, and speech modalities, which brings significant challenges.

Unlike traditional voice cloning (VC) [1, 3] or text-to-speech (TTS) [19, 25, 28, 33] tasks solely depending on the input text for modeling [38], movie dubbing requires maintaining consistency between its variation information (*e.g.*, prosody attributes, duration of each phoneme) and the performances of movie characters. This transforms the modeling of duration and prosody in movie dubbing from a one-to-many to a one-to-one mapping problem while retaining the requirement for accurate pronunciation.

Firstly, regarding duration consistency, movie dubbing needs to match the total duration of the video and further synchronize the lip movements. To this end, previous works incorporate lip motion into the prediction of the duration for each phoneme [4] or utilize an attention-based upsampling module mapping the phoneme sequence to the video frame length to ensure overall duration consistency [5, 12]. However, the former standalone duration prediction for each phoneme overlooks constraints imposed by the total length of the video, while the latter breaks the completeness of a speech unit, leading to unclear pronunciation.

Secondly, in terms of prosody modeling, to reflect the emotions expressed by characters in the video, previous works [4, 12, 23] model the prosody attributes (*e.g.*, pitch and energy) through the emotional expressions of movie characters at the mel-spectrogram frame-level. The excessively fine prosody modeling scale degrades

the model's generalization capability [28], resulting in unstable prosody fluctuations. Furthermore, previous approaches that directly fuse timbre features with phoneme sequences also fail to achieve satisfactory voice cloning effects [4, 5, 12, 23].

Besides, the required audiovisual consistency in movie dubbing emphasizes the comprehensive application of accurate and clear pronunciation. However, movie dubbing datasets are limited in scale (due to copyright issues) and usually involve complicated pronunciation and prosody. The presence of background sounds and ambient noise inherent in movies further increases the difficulty of modeling accurate and clear pronunciation. Existing works [4, 5, 12, 23] try to simultaneously learn pronunciation from scratch and achieve the alignment between dubbing and movie clip solely relying on movie dubbing datasets (as shown in Figure 1 (b) blue arrows), leads to poor pronunciation quality. This demonstrates that the model attempting to leap directly from "baby" to dubber solely depends on movie dubbing datasets is quite difficult and inefficient.

To address the above-mentioned problems, we propose a novel two-stage dubbing method (as shown in Figure 1 (b) green arrows) that better aligns with the practicing process from speaker to dubber. The two-stage framework allows the model to first learn the pronunciation knowledge via multi-task speaker pre-training and then practice synchronizing with movie content to achieve prosody and duration consistency. Specifically, in the first stage, we introduce the multi-task pre-training to learn universal pronunciation representation from a large-scale text-to-speech corpus. It overcomes the limitation of movie dubbing datasets and enhances the pronunciation clarity of dubbing. In the second stage, we design a prosody consistency learning module to bridge the visual emotion features with phoneme-level prosody attributes. Furthermore, timbre-adaptive layer normalization (TALN) is introduced to maintain the timbre consistency with reference audio. Last, we propose a duration consistency reasoning module to predict the optimal monotonic alignment between phoneme-level acoustic features and video frame-level lip motion. This module aims to synchronize dubbing with the movie clip in terms of lip movement and overall duration. Finally, the mel-decoder [34] is used to predict the mel-spectrogram of dubbing, and the vocoder [22] converts the mel-spectrogram into a waveform in the time domain.

The main contributions are summarized as follows:

- We propose a two-stage framework for movie dubbing that enables the model to learn pronunciation from large-scale text-to-speech corpus before practicing in movie dubbing, therefore improving the quality of dubbing pronunciation.
- To better achieve prosody consistency, we propose a prosody consistency learning module to bridge the character's emotion with the phoneme-level prosody attributes and utilize TALN to preserve the vocal timbre of reference audio.
- To achieve duration consistency, we propose a duration consistency reasoning module to predict the optimal alignment between the phoneme-level acoustics features and lip movement in the video.
- Extensive experiments demonstrate that our proposed method outperforms the current state-of-the-art, validating the effectiveness of each module.

## 2 RELATED WORKS

### 2.1 Speech Synthesis

Speech synthesis, or text-to-speech, plays a crucial role in various applications, including virtual assistants, navigation systems, accessibility tools, and entertainment media. In recent years, numerous advancements have been made in speech synthesis. WaveNet [40] and Tacotron [35] architectures demonstrate remarkable performance in generating high-quality and expressive speech in an autoregressive way. Then, the Fastspeech series [33, 34] achieve high-quality and rapid speech synthesis by expanding phoneme sequences to the same length as mel-spectrogram using Montreal forced alignment (MFA) [26] and length regulator. Recently, a proliferation of models based on diffusion [14, 31] or generative flow [9, 20], aiming to generate natural and human-level speech [8, 13, 16, 19, 42]. To further improve the naturalness of synthesized speech, the NaturalSpeech series [17, 36, 38] employ elaborate designed neural speech codecs for speech attribute factorization and a factorized diffusion model to build fine-grained speech attributes. Despite the impressive progress, these methods cannot be directly applied to the V2C task as they lack the required prosody and duration consistency modeling with input move clip.

### 2.2 Pre-training in Text-to-Speech

Pre-trained language models (*e.g.*, BERT [7]) achieve state-of-the-art performance in solving natural language processing tasks. Since training a TTS model is like learning a language from scratch [24], recently many works [11, 15, 24, 29, 45] employ additional BERT encoders to enhance the pronunciation quality and expressiveness of the generated speech. In order to enable the BERT to operate at the phoneme level sequence avoiding the inconsistency between phoneme and character information, MP-BERT [45] merges the phoneme and sup-phoneme sequences into a new sequence, PL-BERT [24] combines whole-word masked phoneme and grapheme prediction as pre-train strategy, both achieve promising results. Pre-training of masked language models provides abundant phoneme-level contextual information which can enhance the naturalness of speech. Therefore, we incorporate this pre-train strategy into the movie dubbing task to enhance the naturalness of dubbing.

### 2.3 Visual Voice Cloning

The V2C task [4] introduces video information as one of the conditions for speech synthesis, requiring the generated dubbing to align with the video content in terms of lip movements, emotions, and duration. This eliminates the one-to-many problem in traditional speech synthesis tasks and significantly increases the complexity of the task. Building on this, a hierarchical prosody learning model is proposed [5], which adapts the prosody of dubbing by learning at three levels: lip movements, character facial information, and global scene information, at the level of video frames. Recent works [12, 23] try to solve the dubbing task using human-face as the timbre identity features using the non-autoregressive model or diffusion model. Although previous works make progress in improving the consistency between dubbing audio and movie clips, the complexity and limited scale of movie dubbing datasets constrain the pronunciation accuracy of learned models.

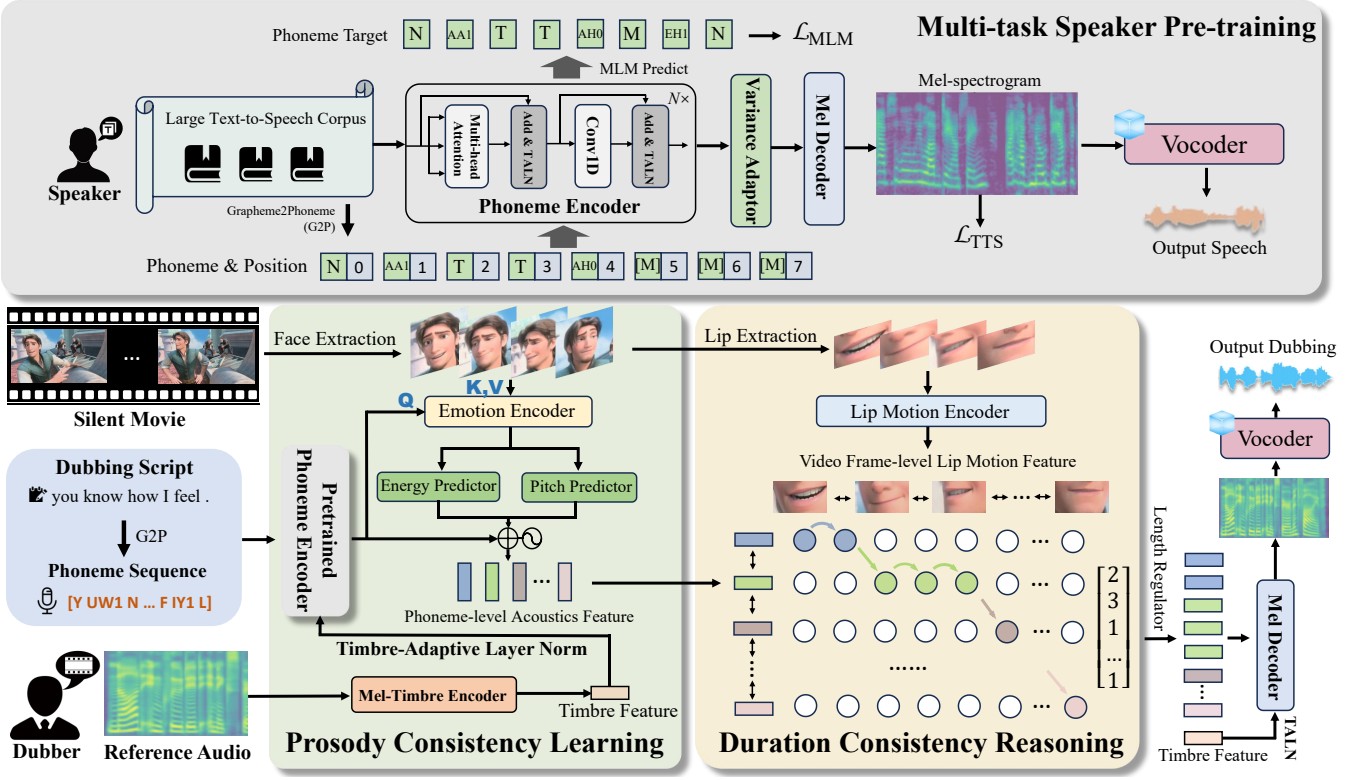

**Figure 2: The main architecture of the proposed two-stage dubbing method. In the first stage, we introduce a multi-task speaker pre-training (Section 3.2) to improve the quality of dubbing pronunciation. In the second stage, we propose the Prosody Consistency Learning (Section 3.3) and Duration Consistency Reasoning (Section 3.4) to improve audiovisual consistency.**

## 3 METHODS

### 3.1 Overview

The target of the overall movie dubbing task is:

$$\tilde{A}_{Dub} = \text{Model}(A_{Ref}, T_s, V_{Ref}), \tag{1}$$

where the $\tilde{A}_{Dub}$ is the generated dubbing and $A_{Ref}, T_s, V_{Ref}$ are the reference audio, scripts, and reference video, *i.e.*, movie clip respectively.

The main architecture of the proposed method is shown in Figure 2. Our method involves two training stages: multi-task speaker pre-training (MTSP) (top) and dubbing training with prosody and duration consistency learning (bottom). In the first stage, we conduct a multi-task speaker pre-training for the phoneme encoder. Since conventional speech and movie dubbing differ significantly in tone and prosody but share common pronunciation semantics, pre-training the phoneme encoder is the optimal choice. Compared to movie dubbing datasets, the recording environment of text-speech corpus is often a noise-free recording studio. The prosody and pronunciation in the corpus are smooth and clear. Therefore it can offer rich pronunciation knowledge for movie dubbing.

In the second stage, we freeze the pre-trained phoneme encoder and train the model using the movie dubbing datasets. We propose the Prosody Consistency Learning (PCL) module and the Duration Consistency Reasoning (DCR) module. These modules enable the model to better practice pronunciation skills thus achieving prosody

and duration consistency with the movie clip. They respectively align the emotion and lip movement of the character with the phoneme-level prosody and duration of the dubbing. Additionally, in the PCL module, we employ a mel-timbre encoder and timbre-adaptive layer normalization (TALN) to extract and integrate the timbre feature of the reference audio, thereby ensuring timbre consistency. We detail each module below.

### 3.2 Multi-task Speaker Pre-training

The pre-training consists of two tasks: the TTS task and the MLM prediction task. The former enables the model learns to extract accurate pronunciation representations from high-quality text-speech corpus and the latter facilitates the model to learn contextual relationships between phonemes and better handle unseen text.

**TTS task.** We adopt an architecture similar to the FastSpeech2 [33] in this stage, where the model primarily consists of three main components for generating the mel-spectrogram of speech, namely the phoneme encoder, variance adaptor, and mel-decoder. Specifically, after the text is converted into phoneme sequence by an open-source grapheme-to-phoneme tool[1] (G2P), it serves as the input to the model. Subsequently, a phoneme encoder composed of stacked Feed-Forward-Transformer (FFT) blocks [34] is utilized to extract phoneme embeddings, and the variance adaptor is employed to model the prosody attributes of speech, namely pitch, energy, and

---

[1]https://github.com/Kyubyong/g2p

the duration of each phoneme. Then transform them into acoustic features with the desired length as the mel-spectrogram as follows:

$$T_p = \text{G2P}(T_o) \in \mathbb{R}^{L_p \times d_p},$$

$$T_e = \text{PhonemeEncoder}(T_p) \in \mathbb{R}^{L_p \times d_m}, \quad (2)$$

$$T_{mel} = \text{VarianceAdaptor}(T_e, D_p, P_p, E_p) \in \mathbb{R}^{L_{mel} \times d_m},$$

where $T_o$, $T_p$, and $T_e$ are the original text, phoneme sequence, and phoneme embedding, respectively. $L_p$ and $L_{mel}$ denote the length of the phoneme sequence and desired mel-spectrogram, respectively. $d_m$ denotes the hidden size of the model. $D_p$, $P_p$, and $E_p$ denote the duration and prosody attributes of each phoneme, namely pitch and energy. $T_{mel}$ denotes the expended phoneme-level acoustic sequence with prosody attributes.

It is worth noting that in the original TTS task, the duration, pitch, and energy of each phoneme are predicted by the variance adaptor directly from the phoneme sequence. However, in the movie dubbing task, the duration and the prosody attributes of dubbing need to be bridged with visual information, such as the lip movements and emotional expressions of characters. Since no matching visual information is available in the text-speech corpus, we directly use the ground truth duration, pitch, and energy information in the TTS pre-training task. Furthermore, to enable the model to better adapt to a multi-speaker environment during the pre-training phase, we also employ the mel-timbre encoder and TALN at this stage. We provide a detailed explanation of these components in Section 3.3. The omission here does not affect the description of the pre-training process.

Then, we predict the mel-spectrogram of target speech and calculate the loss of this task:

$$\widetilde{M} = \text{MelDecoder}(T_{mel}),$$

$$\mathcal{L}_{TTS} = \frac{1}{L_{mel}} \sum_{t=0}^{L_{mel}-1} \left\| M_t - \widetilde{M}_t \right\|, \quad (3)$$

where the mel-decoder is composed of stacked FFT blocks [34] similar to the phoneme encoder, $M$ and $\tilde{M}$ denote the ground truth mel-spectrogram and the predicted one, respectively. Subscript $t$ denotes the $t$-th frame.

**MLM prediction task.** The MLM prediction task enables the model to enhance the naturalness of dubbing audio by learning phoneme contextual relationships in a self-supervised manner.

Specifically, we convert the text in the corpus into phoneme sequences and then randomly mask the sequences with a predetermined masking ratio. After inputting the phoneme sequences into the phoneme encoder, we predict the masked input phoneme tokens from the hidden states of the last layer using a linear projection along with a softmax function. The loss function is the cross-entropy loss commonly used for multi-class prediction. The training target of the MLM prediction task is:

$$\mathcal{L}_{MLM} = CE(\text{PhonemeEncoder}(T_{mask}), T_p), \quad (4)$$

where the $T_{mask}$ is the masked phoneme sequence and $CE(\cdot)$ denotes the cross-entropy loss.

**Loss function of the first stage.** During the multi-task speaker pre-training, the TTS task and MLM prediction task are conducted separately. The masking in the MLM prediction task does not affect

the encoding of phonemes in the TTS task. The total loss for the first stage is as follows:

$$\mathcal{L}_{MTSP} = \alpha_1 \cdot \mathcal{L}_{TTS} + \alpha_2 \cdot \mathcal{L}_{MLM}, \quad (5)$$

where the $\alpha_1$ and $\alpha_2$ are pre-defined hyper parameters.

## 3.3 Prosody Consistency Learning

The Prosody Consistency Learning (PCL) module consists of two parts: 1) Bridging the emotional states of characters in movie clips with the phoneme-level prosody attributes of dubbing and 2) accurately replicating the timbre of the reference audio.

**Bridging emotion with dubbing.** In movie dubbing, the consistency between the dubbing's prosody and the characters' emotions is crucial. A professional dubber always observes the characters' facial expressions in the movie to adjust their pronunciation prosody. To enable the model to achieve this, we model the phoneme-level prosody attributes pitch and energy based on the facial information of the characters in the video.

Following [5], we exact the face region of each frame of movie clip $V_{face}$ via S$^3$FD face detection [46], then utilize an emotion face-alignment network (EmoFAN) [39] to encode the face region to emotion features $V_{emo}$:

$$V_{face} = \text{S}^3\text{FD}(V_{Ref}) \in \mathbb{R}^{L_v \times H_{face} \times W_{face} \times C},$$

$$V_{emo} = \text{EmoFAN}(V_{face}) \in \mathbb{R}^{L_v \times d_m}, \quad (6)$$

where the $L_v$ denotes the frame number of the reference video.

To model the pitch and energy of each phoneme based on the emotion from video frames, we utilize multi-head cross-modal attention to bridge the character's emotion features with prosody attributes of each phoneme:

$$\xi_{pho,pitch}^k = softmax(\frac{Q^\top K_p}{\sqrt{d_m}})V_p \in \mathbb{R}^{L_p \times \frac{d_m}{n\_head}},$$

$$\xi_{pho,enrgy}^k = softmax(\frac{Q^\top K_e}{\sqrt{d_m}})V_e \in \mathbb{R}^{L_p \times \frac{d_m}{n\_head}}, \quad (7)$$

$$Q = W_j^Q T_e^\top, K_p = W_j^{K_p} V_{emo}^\top, V_p = W_j^{V_p} V_{emo}^\top,$$

$$K_e = W_j^{K_e} V_{emo}^\top, V_e = W_j^{V_e} V_{emo}^\top,$$

where $k$ denotes $k$-th head's output, $W_j^*$ are learnable parameter matrix, $\xi_{pho,pitch}$ and $\xi_{pho,energy}$ are phoneme-level prosody feature for pitch and energy, respectively. After bridging the prosody of dubbing with the character's emotion, we predict the pitch energy of each phoneme using the pitch and energy predictor and transfer them to pitch and energy embedding then add to the phoneme sequence:

$$\widetilde{P}_{pho}, \widetilde{E}_{pho} = \text{Predictor}(\xi_{pho,pitch}, \xi_{pho,energy}) \in \mathbb{N}^{L_p},$$

$$T_a = T_e + \text{PitchEmb}(\widetilde{P}_{pho}) + \text{EnergyEmb}(\widetilde{E}_{pho}), \quad (8)$$

where $\widetilde{P}_{pho}$ and $\widetilde{E}_{pho}$ are the predicted pitch and energy and $T_a \in \mathbb{R}^{L_p \times d_m}$ are the phoneme-length acoustics feature with prosody attributes.

**Timbre consistency.** To keep the timbre consistency within the same character and ensure a coherent and consistent understanding of the characters by the audience, we propose a mel-timbre encoder to extract the timbre feature from reference audio.

The mel-timbre encoder comprises three parts: (1) a fully connected layer to transform each frame of the mel-spectrogram into the hidden sequence of acoustics features, (2) a temporal processing module to capture contextual information using a stacked 1-D convolutional neural network, and (3) a multi-head attention module to aggregate global information of vocal timbre. Due to the issue of poor generalization of learnable timbre encoder in movie dubbing datasets, we pre-train the mel-timbre encoder on the multi-speaker text-speech corpus [44] using GE2E loss [41] and froze its weights during both stages of training unlike [28].

Inspired by the style transfer in the image and speech generation domain [18, 28], we utilize a Timbre-Adaptive Layer Norm to fuse the timbre feature into phoneme encoding and mel-spectrogram generation by predicting the gain and bias of the input vector sequence giving the timbre feature:

$$\text{TALN}(x, E_{timbre}) = gain(E_{timbre})\frac{x - \mu}{\sigma} + bias(E_{timbre}), \quad (9)$$

where $x$ and $E_{timbre}$ are the input sequence and timbre feature, $\mu$, $\sigma$ are the mean and variance of $x$, $gain(\cdot)$ and $bias(\cdot)$ are the prediction function of gain and bias, respectively. The proposed TALN is applied in every FFT block of both the phoneme encoder and mel-decoder to integrate the vocal timbre feature.

## 3.4 Duration Consistency Reasoning

The proposed Duration Consistency Reasoning module (DCR) contains two steps: (1) reasoning the phoneme-lip alignment and (2) expanding to the length of desired mel-spectrogram.

**Reasoning the phoneme-lip alignment.** To exact lip motion features from the reference video, we first extract the lip region $V_{Lip} \in \mathbb{R}^{L_v \times H_{Lip} \times W_{Lip} \times C}$ from the video, then exploit a lip motion encoder to obtain the lip movement representation $E_{Lip} \in \mathbb{R}^{L_v \times d_{model}}$.

After obtaining the lip motion features from the movie clip, we can calculate the upper triangular similarity matrix between phoneme-level acoustic features and lip-motion features:

$$S_{pho,lip} = \begin{bmatrix} s_{11} & s_{12} & \cdots & s_{1L_v} \\ -inf & s_{22} & \cdots & s_{2L_v} \\ \vdots & \vdots & \ddots & \vdots \\ -inf & -inf & \cdots & s_{L_pL_v} \end{bmatrix}, s_{i,j} = Similarity(T_a^i, E_{lip}^j),$$

$$(10)$$

where the $s_{i,j}$ is the cosine similarity between $i$-th phoneme-level acoustic feature and $j$-th frame's lip motion feature.

The alignment between lip phoneme and lip motion should satisfy surjectivity, which means the alignment path must begin at $s_{11}$ and end at $s_{L_pL_v}$. Additionally, this alignment also needs to satisfy monotonicity, each step of the alignment path can only move to the right or diagonally down to the right, thus we can use dynamic programming algorithms to find the optimal alignment:

$$A_{i,j} = \begin{cases} None, & \text{if } i > j \text{ or } j - i < L_v - L_p \\ max(A_{i-1,j}, A_{i-1,j-1}) + s_{i,j}, & \text{otherwise} \end{cases},$$

$$(11)$$

$$A^* = \underset{j}{\arg\max} \, A_{ij}, \quad (12)$$

where $A$ is the alignment matrix and $A^*$ is the optimal alignment between phoneme-level acoustic and lip-motion features which simultaneously satisfy monotonicity and surjectivity.

**Expanding to the mel-length.** The duration of the video corresponds strictly to the duration of the audio, hence, the number of video frames corresponds one-to-one with the length of the corresponding mel-spectrogram, maintaining a fixed ratio relationship [5, 12]:

$$n = \frac{L_{mel}}{L_v} = \frac{sr/hs}{FPS} \in \mathbb{N}^+, \quad (13)$$

where the $sr$ and $hs$ are the sampling rate and hop size when processing the audio and $FPS$ denotes the Frames Per Second of the video. With this fixed coefficient $n$, we can extend the obtained phoneme-lip alignment to phoneme-mel-spectrogram alignment, acquiring the duration of each phoneme. Then we expand the acoustics feature sequence of phoneme length to the required mel-spectrogram length using length regulator [34] for mel-spectrogram generation and audio generation:

$$\tilde{A}_{Dub} = \text{Vocoder}(\text{MelDecoder}(\text{LR}(T_a, A^* \times n), E_{timbre})), \quad (14)$$

where the $\text{LR}(\cdot)$ denotes the length regulator.

## 3.5 Loss Function of Second Stage

The total loss function of this training stage is:

$$\mathcal{L}_{total} = \lambda_1 \mathcal{L}_{mel} + \lambda_2 \mathcal{L}_{pitch} + \lambda_3 \mathcal{L}_{enrgy} + \lambda_4 \mathcal{L}_{align}, \quad (15)$$

where $\lambda_*$ are pre-defined hyper-parameters, $\mathcal{L}_{mel}$, $\mathcal{L}_{pitch}$, and $\mathcal{L}_{energy}$ are the L1 Loss to the mel-spectrogram and the prediction of the pitch and energy of each phoneme respectively. For the alignment in duration consistency reasoning, since the dynamic programming algorithm does not have any learnable parameters, we optimize the lip motion encoder using ground truth alignment, gradually making the similarity matrix approach the ideal alignment. For detailed information refer to Appendix.

## 4 EXPERIMENTS

We primarily evaluate our method on two dubbing datasets: V2C-Animation [4] and GRID [6]. Here, we present our experimental results from various aspects, including dataset description, implementation details, evaluation metrics, qualitative and quantitative analysis, as well as ablation studies.

## 4.1 Datasets

**V2C-Animation dataset [4]** is currently the only publicly available movie dubbing dataset. Specifically, it consists of 10,217 video-audio-text triplets cropped from 26 Disney animated movies, totaling 153 different speakers, with complete speaker and emotion annotations. The dataset extracts the middle audio track from movie clips as dubbing data but still exists some environmental sounds and noise that cannot be eliminated. Due to the authenticity of this dataset which is from publicly available movies, it's the most challenging dataset for the V2C task currently. Therefore, our main experimental results are validated on this dataset.

**GRID dataset [6]** is a basic benchmark for multi-speaker dubbing. The whole dataset has 33 speakers, each with 1,000 short English samples. All participants are recorded in a noise-free studio with a unified screen background. The V2C-Animation dataset and the

**Table 1: Results on V2C-Animation benchmark. The method with "*" refers to a variant taking video embedding as an additional input following [4]. For the Dub 1.0 setting, we use the ground truth audio as reference audio, for the Dub 2.0 setting, we use the non-ground truth audio from the same speaker within the dataset as the reference audio which is more aligned with practical usage in dubbing. The same setup is applied to the GRID benchmark.**

| Setting | | Dub 1.0 | | | | | Dub 2.0 | | | | |
| --- | --- | --- | --- | --- | --- | --- | --- | --- | --- | --- | --- |
| Methods | Visual | SECS (%) ↑ | WER (%) ↓ | EMO-ACC (%) ↑ | MCD-DTW ↓ | MCD-DTW-SL ↓ | SECS (%) ↑ | WER (%) ↓ | EMO-ACC (%) ↑ | MCD-DTW ↓ | MCD-DTW-SL ↓ |
| GT | - | 100.00 | 25.55 | 99.96 | 0.00 | 0.00 | 100.00 | 22.55 | 99.96 | 0.00 | 0.00 |
| GT Mel + Vocoder | - | 96.96 | 24.40 | 97.09 | 3.77 | 3.80 | 96.96 | 24.40 | 97.09 | 3.77 | 3.80 |
| Fastspeech2 [33] | X | 24.87 | 34.48 | 42.21 | 11.20 | 14.48 | 24.17 | 35.08 | 42.21 | 11.20 | 14.48 |
| StyleSpeech [28] | X | 54.99 | 106.73 | 44.12 | 11.50 | 15.10 | 75.66 | 76.58 | 41.55 | 11.56 | 15.10 |
| Zero-shot TTS [47] | X | 48.98 | 68.81 | 42.75 | 9.98 | 12.51 | 47.79 | 58.82 | 39.11 | 10.68 | 13.52 |
| Matcha-TTS [27] | X | 16.88 | 89.50 | 41.66 | 10.59 | 10.79 | 16.88 | 89.50 | 41.66 | 10.59 | 10.79 |
| Fastspeech2* [33] | ✓ | 25.47 | 33.53 | 42.39 | 11.35 | 14.73 | 25.47 | 34.08 | 42.39 | 11.35 | 14.73 |
| StyleSpeech* [28] | ✓ | 42.53 | 108.00 | 42.53 | 11.62 | 14.23 | 75.67 | 82.48 | 42.57 | 11.58 | 14.73 |
| Zero-shot TTS* [47] | ✓ | 48.93 | 68.05 | 43.97 | 10.03 | 12.01 | 47.55 | 58.81 | 39.30 | 10.76 | 13.66 |
| V2C-Net [4] | ✓ | 40.61 | 73.08 | 43.08 | 14.12 | 18.49 | 34.07 | 61.61 | 41.01 | 14.58 | 18.73 |
| HPMDubbing [5] | ✓ | 53.76 | 164.16 | 46.61 | 11.12 | 11.22 | 31.42 | 171.03 | **43.97** | 11.88 | 11.98 |
| Face-TTS [23] | ✓ | 52.81 | 201.13 | 44.04 | 13.44 | 26.94 | 51.98 | 200.18 | 43.56 | 13.78 | 28.03 |
| Ours | ✓ | **81.50** | **17.51** | **46.80** | **9.46** | **9.65** | **79.86** | **17.33** | 43.66 | **10.64** | **10.84** |

GRID dataset correspond respectively to dubbing for animated films and live-action recording, covering a wide range of application scenarios.

**LibriTTS dataset [44]** is a multi-speaker English corpus derived from LibriSpeech [30]. LibriTTS comprises 110 hours of audio from 1,141 speakers along with their corresponding text transcripts. It filters out mismatched or significantly noisy samples found in LibriSpeech and is widely used in speech synthesis or automatic speech recognition (ASR) tasks. Due to its advantages in speech quality and scale as well as the similar multi-speaker settings, we employ LibriTTS as our speaker pre-training dataset in the first stage.

## 4.2 Implementation Details

The video frames are sampled at 25 FPS and all audios are resampled to 22.05kHz. For all audio data, we convert the raw waveform into mel-spectrograms following [4, 5, 28, 33, 34] with FFT size of 1024, hop size of 256, window size of 1024, and frequency bins of 80. The ground truth of phoneme duration is extracted by Montreal Forced Aligner [26]. We use continuous wavelet transform (CWT) to decompose the continuous pitch series into pitch spectrograms to get the phoneme-level pitch [10, 37]. For energy extraction, we compute the mean L2-norm of the amplitude of each short-time Fourier transform (STFT) frame within a phoneme duration [33].

For the MLM prediction task of the multi-task speaker pre-training in the first stage, we use a 15% random masking rate and train with a batch size of 16 for 100 epochs together with the TTS task on the LibriTTS-Clean-100 dataset. The weight in Equation 5 are set to $\alpha_1 = 1$, $\alpha_2 = 0.1$. In the second stage of training, we freeze the pre-trained phoneme encoder and train the rest part of the model on the dubbing datasets. We use a pre-trained universal speaker version of HiFi-GAN [22] as vocoder to convert the mel-spectrogram to time-domain waveforms. The weight in Equation 15 are set to $\lambda_1 = 1$, $\lambda_2 = 0.1$, $\lambda_3 = 0.1$, $\lambda_4 = 0.2$. An Adam [21] with $\beta_1 = 0.9$, $\beta_2 = 0.98$, $\epsilon = 10^{-9}$ is used as the optimizer in both the training stages. The learning rate is set to 0.00625. Both training and inference are implemented with PyTorch on a GeForce RTX 4090 GPU. For a fair comparison, all comparison models are re-trained on the same dataset. For more details of training and implementation please refer to Appendix.

## 4.3 Evaluation Metrics

**Objective metrics**. To measure the difference between the generated dubbing and the ground truth, we adopt the Mel Cepstral Distortion Dynamic Time Warping (MCD-DTW) metric following [4]. To further assess the duration consistency between the generated dubbing and the video, we utilize the MCD-DTW-SL metric which adjusts the weights based on duration consistency [4]. Furthermore, to evaluate the timbre consistency between the generated dubbing and the reference audio, we employ the speaker encoder cosine similarity (SECS) following [2, 3] to compute the similarity of speaker identity. To assess the pronunciation quality of the generated dubbing, we utilize the state-of-the-art automatic speech recognition (ASR) model whisper[2] [32] from OpenAI for dubbing recognition and computing the word error rate (WER) [3] against the script to evaluate the accuracy of the generated dubbing. In addition, we utilize a speech emotion recognition model [43] to evaluate the emotion accuracy (EMO-ACC) of the generated dubbing (For the V2C-Animation benchmark only because there is no emotion label in GRID dataset).

**Subjective metrics**. For subjective evaluation, we conduct human evaluations of mean opinion score (MOS) in aspects of naturalness (NMOS) and similarity (SMOS). Both metrics are rated on a 1-to-5 scale and reported with the 95% confidence intervals (CI). Following the settings in [4, 5], all participants are asked to assess the dubbing quality of 30 randomly selected audio samples from each test set. In addition to SMOS and NMOS, we also utilize Comparative MOS (CMOS) with 7 points (from -3 to 3) to compare different models.

## 4.4 Comparison with SOTA

**Results on V2C-Animation benchmark.** As shown in Table 1, our model achieves the best performance across almost all metrics, except for the EMO-ACC metric in the dub 2.0 setting, where it slightly lagged behind the current SOTA dubbing model HPM-Dubbing [5]. In both dub settings, our method achieves the best performance on SECS. It indicates that our method performs better in extracting and cloning vocal timbre. Since the V2C-Animation

---

[2]https://huggingface.co/openai/whisper-large
[3]https://github.com/jitsi/jiwer

**Table 2: Results on GRID benchmark with the same dub setting as the V2C-Animation benchmark.**

| Setting | | Dub 1.0 | | | | Dub 2.0 | | | |
|---|---|---|---|---|---|---|---|---|---|
| Methods | Visual | SECS (%) ↑ | WER (%) ↓ | MCD-DTW ↓ | MCD-DTW-SL ↓ | SECS (%) ↑ | WER (%) ↓ | MCD-DTW ↓ | MCD-DTW-SL ↓ |
| GT | - | 100.00 | 22.41 | 0.00 | 0.00 | 100.00 | 22.41 | 0.00 | 0.00 |
| GT Mel + Vocoder | - | 97.57 | 21.41 | 4.10 | 4.15 | 97.57 | 21.41 | 4.10 | 4.15 |
| Fastspeech2 [33] | X | 47.41 | 19.05 | 7.67 | 8.43 | 47.41 | 19.05 | 7.67 | 8.43 |
| StyleSpeech [28] | X | 91.06 | 24.83 | 5.87 | 5.98 | 74.15 | 21.42 | 7.02 | 7.95 |
| Zero-shot TTS [47] | X | 86.54 | 19.13 | 5.71 | 5.99 | 82.25 | 19.35 | 6.21 | 6.76 |
| Fastspeech2* [33] | ✓ | 25.47 | 19.61 | 11.35 | 14.73 | 59.58 | 19.61 | 7.24 | 7.95 |
| StyleSpeech* [28] | ✓ | 90.04 | 22.62 | 5.74 | 5.88 | 59.58 | 19.82 | 7.01 | 7.82 |
| Zero-shot TTS* [47] | ✓ | 85.93 | 20.05 | 5.75 | 6.40 | 81.34 | 21.05 | 6.27 | 7.29 |
| V2C-Net [4] | ✓ | 80.98 | 47.82 | 6.79 | 7.23 | 71.51 | 49.09 | 7.29 | 7.86 |
| HPMDubbing [5] | ✓ | 85.11 | 45.51 | 6.49 | 6.78 | 71.99 | 44.15 | 6.79 | 7.09 |
| Face-TTS [23] | ✓ | 82.97 | 44.37 | 7.44 | 8.16 | 34.14 | 39.05 | 7.77 | 8.59 |
| Ours | ✓ | **94.50** | **17.07** | **5.34** | **5.45** | **85.76** | **17.42** | **6.17** | **6.43** |

**Table 3: Subjective evaluation on V2C-Animation and GRID benchmarks.**

| Dataset | V2C-Animation | | | GRID | | |
|---|---|---|---|---|---|---|
| Methods | NMOS ↑ | SMOS ↑ | CMOS ↑ | NMOS ↑ | SMOS ↑ | CMOS ↑ |
| GT | 4.52±0.13 | - | +0.23 | 4.69±0.07 | - | +0.14 |
| GT Mel + Vocoder | 4.39±0.16 | 4.41±0.18 | +0.21 | 4.66±0.08 | 4.53±0.10 | +0.16 |
| Fastspeech2 [33] | 3.27±0.12 | 2.94±0.18 | -0.29 | 3.37±0.14 | 3.09±0.11 | -0.26 |
| StyleSpeech [28] | 3.34±0.13 | 3.37±0.14 | -0.22 | 3.56±0.14 | 3.60±0.19 | -0.25 |
| Zero-shot TTS [47] | 3.38±0.14 | 3.50±0.19 | -0.26 | 3.57±0.12 | 3.54±0.13 | -0.23 |
| Fastspeech2* [33] | 3.29±0.10 | 2.90±0.21 | -0.27 | 3.31±0.12 | 3.04±0.17 | -0.26 |
| StyleSpeech* [28] | 3.31±0.21 | 3.35±0.12 | -0.20 | 3.50±0.10 | 3.58±0.11 | -0.24 |
| Zero-shot TTS* [47] | 3.40±0.12 | 3.47±0.18 | -0.24 | 3.58±0.21 | 3.52±0.15 | -0.21 |
| V2C-Net [4] | 3.54±0.16 | 3.51±0.18 | -0.21 | 3.62±0.06 | 3.67±0.11 | -0.19 |
| HPMDubbing [5] | 3.57±0.17 | 3.54±0.12 | -0.18 | 3.77±0.20 | 3.74±0.13 | -0.14 |
| Face-TTS [23] | 3.18±0.13 | 3.24±0.16 | -0.37 | 3.39±0.21 | 3.32±0.17 | -0.32 |
| Ours | **3.92±0.19** | **3.87±0.14** | **0.00** | **4.03±0.09** | **4.05±0.11** | **0.00** |

**Table 4: Results on zero-shot test.**

| Method | Visual | SECS ↑ | WER ↓ | NMOS ↑ | SMOS ↑ | CMOS ↑ |
|---|---|---|---|---|---|---|
| FastSpeech2 [33] | X | 21.11 | 27.73 | 3.34±0.10 | 3.14±0.16 | -0.25 |
| StyleSpeech [28] | X | 55.81 | 93.40 | 3.49±0.17 | 3.52±0.21 | -0.19 |
| Zero-shot TTS [47] | X | 57.23 | 31.47 | 3.53±0.16 | 3.56±0.11 | -0.18 |
| FastSpeech2* [33] | ✓ | 26.79 | 30.27 | 3.31±0.07 | 3.19±0.13 | -0.24 |
| StyleSpeech* [28] | ✓ | 58.71 | 105.64 | 3.51±0.12 | 3.52±0.23 | -0.21 |
| Zero-shot TTS* [47] | ✓ | 61.12 | 35.10 | 3.54±0.21 | 3.57±0.12 | -0.16 |
| V2C-Net [4] | ✓ | 39.43 | 143.54 | 3.61±0.22 | 3.64±0.17 | -0.14 |
| HPMDubbing [5] | ✓ | 49.31 | 106.45 | 3.62±0.16 | 3.61±0.23 | -0.11 |
| FaceTTS [23] | ✓ | 33.80 | 231.63 | 3.46±0.09 | 3.51±0.17 | -0.29 |
| Ours | ✓ | **73.44** | **16.05** | **3.85±0.12** | **3.87±0.09** | **0.0** |

dataset is derived from real movie dubbing clips, its samples involve complex pronunciation and prosody variations, which increases the difficulty for the model to learn accurate pronunciation from them. Previous methods fail to perform accurate pronunciation, as reflected in the high WER values. However, our model achieves pronunciation accuracy significantly better than other models with an absolute margin from 16.52% to 183.62%. While ensuring pronunciation accuracy, our model also achieves the lowest MCD-DTW and MCD-DTW-SL, indicating smaller discrepancies compare to ground truth dubbing and better duration consistency.

In the subjective evaluation, we randomly select 15 samples from the generated dubbing of each dub setting for human study. Table 3 shows the results. All objective evaluations of our model in the V2C benchmark achieve the highest scores. It demonstrates that our model can generate dubbing closer to realistic dubbing in both naturalness and similarity. In addition, our model outperforms the

current state-of-the-art dubbing model [5] in CMOS evaluation with a margin of +0.18.

**Results on GRID benchmark.** As shown in Table 2, our model achieves the best performance across all metrics on the GRID benchmark. Unlike V2C-Animation, samples in GRID are recorded in a studio environment which does not involve exaggerated prosody variation and background noise. Therefore, the pronunciation accuracy of all comparison methods is generally better on the GRID benchmark compared to the V2C-Animation benchmark. Nevertheless, our model still achieves the best pronunciation clarity (see WER) and the best SECS in both dub settings. The lowest MCD-DTW and MCD-DTW-SL demonstrate the ability of our model to generate dubbing closer to ground truth. Moreover, our model also achieves the highest scores in naturalness and similarity in subjective evaluations and surpasses other models in comparative evaluations with a margin of +0.14.

**Results of Zero-shot test.** In addition to the evaluation on two benchmarks, we also conduct a zero-shot experiment to verify the robustness of our method. In the zero-shot experiment, we utilize scripts and movie clips from the V2C-Animation dataset and reference audio from the GRID dataset (*i.e.*, out of domain) to simulate the application of generating customized dubbed videos in real-world scenarios. Due to the absence of corresponding ground truth in this test, we only calculate WER and SECS for objective evaluation to assess pronunciation quality and timbre consistency.

As shown in Table 4, our model achieves the best pronunciation clarity and timbre consistency in the zero-shot test. It demonstrates that our method can maintain stable dubbing synthesis when facing reference audio from out-of-domain sources. The subjective evaluation results also demonstrate the superiority of our approach over other models. Our model surpasses the state-of-the-art model in naturalness, similarity, and comparison tests.

## 4.5 Qualitative Analysis

We visualize the mel-spectrograms of ground-truth and synthesized audios by our model and the other two state-of-the-art methods in Figure 3. The red and white bounding boxes represent regions where different models exhibit significant differences in duration consistency and pronunciation details compared to the ground truth. Through the observation of the red bounding box, it is evident that our model outperforms others in maintaining duration consistency.

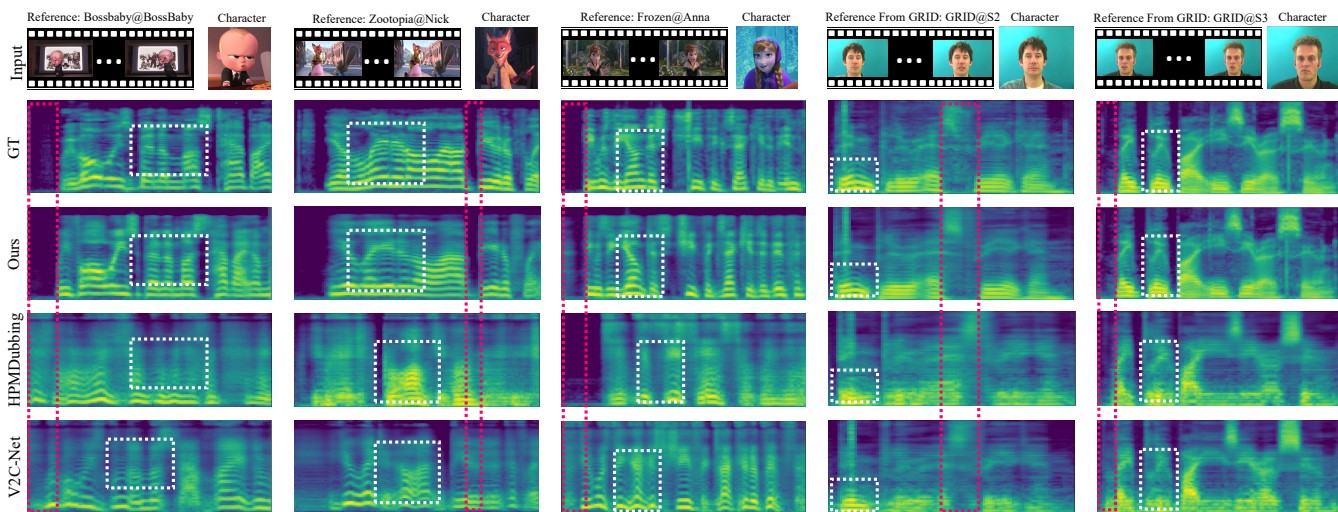

**Figure 3: The visualization of the mel-spectrograms from ground truth and synthesized audios by different models. The red and white bounding boxes highlight regions where different models exhibit significant differences in duration pausing and pronunciation details.**

The phoneme and pause durations are notably closer to the ground truth dubbing. This phenomenon is more pronounced in the V2C-Animation benchmark due to its complex speaking speed variation. Additionally, from the clearer spectrum lines in the white bounding box, it can be observed that the dubbing generated by our model exhibits clearer and more natural pronunciation details.

## 4.6 Ablation Studies

To further investigate the specific effects of each module in our proposed method, we conduct ablation studies on the Dub 1.0 setting on the V2C-Animation benchmark.

**Effectiveness of PCL.** The results are presented in Row A-C of Table 5. We find that the incorporation of TALN enables the model to better clone the timbre of the reference audio. Furthermore, since TALN does not directly affect phoneme features as [4], the pronunciation accuracy of the model is also improved. The PCL module utilizes emotional information of characters to model the phoneme-level prosody attributes of dubbing, enhancing the model's ability to capture character emotions. Compared to mel-spectrogram frame-level prosody modeling (Row A-B), the phoneme-level modeling used in PCL improves the EMO-ACC. It demonstrates a better alignment between dubbing prosody and the emotion of the character. Besides, the integration of PCL also reduces the distance between the generated mel-spectrogram and the ground truth, demonstrating its effectiveness in improving dubbing quality.

**Effectiveness of MTSP.** The results are presented in Row D-F of Table 5. The results indicate that the TTS pre-training task can significantly enhance the pronunciation quality, and improve the WER by an absolute margin of 35.02%. The integration of the TTS pre-training task enables our model to generate dubbing with pronunciation accuracy even reaching the level of ground truth. (We do not fine-tune the ASR model on the movie dubbing datasets, so it may even outperform the ground truth.) Additionally, although the MLM pre-training task does not contribute to pronunciation quality improvement as noticeably as the TTS pre-training task,

**Table 5: Results of ablation study**

| # | Method | SECS ↑ | WER ↓ | EMO-ACC ↑ | MCD-DTW ↓ | MCD-DTW-SL ↓ |
|---|--------|--------|-------|-----------|-----------|--------------|
| - | GT | 100.00 | 22.55 | 99.96 | 0.00 | 0.00 |
| - | GT Mel+ Vocoder | 96.96 | 24.40 | 97.09 | 3.77 | 3.80 |
| A | Baseline [4] | 40.61 | 73.08 | 43.08 | 14.12 | 18.49 |
| B | A+TALN | 76.54 | 65.42 | 43.65 | 11.04 | 12.35 |
| C | A+PCL | 78.19 | 60.81 | 46.50 | 10.06 | 11.90 |
| D | C+MLM Pretrain | 79.03 | 40.28 | 46.67 | 10.84 | 12.39 |
| E | C+TTS Pretrain | 80.00 | 25.79 | 46.17 | 9.72 | 12.24 |
| F | C+MTSP | 80.87 | 20.46 | 46.43 | 9.77 | 11.65 |
| G | DCR *v.s.* Duration Prediction | 81.18 | 21.46 | 46.53 | 9.65 | 11.53 |
| H | Full Model | **81.50** | **17.51** | **46.80** | **9.46** | **9.65** |

their combination can lead to better performance improvements for the model. The improvement in pronunciation quality also reduces the deviation between synthesized and ground truth dubbing.

**Duration Consistency Reasoning *v.s.* Duration Prediction.** We compare the performance of our model using duration prediction (Row G) and the proposed DCR (Row H) respectively. The proposed duration consistency reasoning method considers the relationships between dubbing-lip motion and video-audio synchronization. Unlike the duration prediction method, it does not predict the duration of each phoneme separately, thus avoiding inconsistencies in the overall duration compared to the video. The drops on MCD-DTW-SL demonstrate the effectiveness of DCR.

## 5 CONCLUSIONS

In this work, we propose a two-stage dubbing method to improve the pronunciation quality of dubbing and achieve better audiovisual consistency. In the first stage, we conduct multi-task pre-training on a large-scale text-speech corpus to enable the model to learn universal pronunciation knowledge. In the second stage, the proposed prosody and duration consistency learning module bridges the phoneme-level duration and prosody of dubbing with the movie clip, towards consistency in both aspects. Experimental results demonstrate our method outperforms the current state-of-the-art (SOTA) on two primary benchmarks across multiple settings, validating the effectiveness of our method.

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
