# OpenReview forum: "From Speaker to Dubber: Movie Dubbing with Prosody and Duration Consistency Learning"
_acmmm.org/ACMMM/2024/Conference — MM2024 Oral_

### Official Review · Reviewer_Ensz · 2024-05-17

**Rating:** 5
**Confidence:** 3

**Summary:**

This paper attempts the task of movie dubbing. As there is limited data for this task, the authors propose a two-staged pipeline as the solution to leverage the abundance of TTS data. In the first stage, the authors pre-train a phoneme encoder from large-scale high-quality text-speech corpus.  While in the second stage, the authors introduce the prosody consistency learning and duration consistency reasoning to better align the generated speech with prosody, timbre, duration of movie characters. Afterwards, the authors evaluate the proposed methods on two benchmark datasets, V2C-Animation and GRID with both subjective and objective metrics. Both qualitative and quantitative results demonstrate the proposed methods achieve the state-of-the-art performance. Besides, the authors conduct ablation study to show the effectiveness of proposed sub-modules.

**Strengths:**

1. The solutions are well-motivated. Considering that the movie dubbing data is limited, the authors propose to leverage the text-speech corpus. Besides, several sub-modules are proposed to maintain the consistency of prosody, durations, and timbre.
2. The performance, especially in terms of quantitative aspect, is strong. In Table 1-4, the authors have shown the state-of-the-art performance of proposed approach. Through the provided generated samples, the proposed approach does perform better than baselines.
3. The experiments are comprehensive. The authors conduct experiments on two benchmark datasets with different settings under various evaluation metrics. Moreover, detailed ablation studies have been conducted as shown in Table 5.
4. The writing is clear and smooth to the readers and the pipeline is well illustrated. Although I have some concerns about (a) why the proposed method has lower WER than GT methods and (b) how to ensure the output from energy predictor and pitch predictor learn the actual energy and pitch and (c) where the performance gain on timbre consistency comes from, I am glad I have found the answers from the later sections.

**Limitations:**

1. Although the quantitative results are strong in the main paper, after I listen to the provided samples, I still think the generated samples by the proposed method have identity/timbre shift compared to the ground-truth.  The improvement on emotion accuracy is limited, which also is reflected through the generated samples.  Therefore, I suggest to include the discussions on limitations of proposed method.
2. As there are many modules in the proposed architecture, I think it is better to clear indicate which module is trainable or frozen in Figure 2 as well as the main text. For example, Mel-timbre encoder is frozen in Prosody consistency learning but not highlighted in Figure 2.
3. The details on train/valid/test splits of benchmark datasets are absent.

**Suitability:**

3

---

### Official Review · Reviewer_Kg6k · 2024-05-21

**Rating:** 6
**Confidence:** 4

**Summary:**

This paper tackles the challenge of Movie Dubbing, which aims to generate speech from scripts that aligns temporally and emotionally with a movie clip while maintaining the speaker's voice from a brief reference audio. The authors identify the complexities arising from diverse emotions, pacing, and background noise, coupled with limited data availability due to copyright restrictions. To overcome these hurdles, a two-stage dubbing method is proposed.
The first stage focuses on learning clear phoneme pronunciations using a multi-task approach and a large-scale text-to-speech corpus. In the second stage, a prosody consistency learning module bridges emotional expression with phoneme-level dubbing prosody (pitch and energy). Additionally, a duration consistency reasoning module aligns dubbing duration with lip movements. Experiments on benchmark datasets demonstrate the superiority of this method over existing state-of-the-art approaches. The authors plan to release their source code and model checkpoints for public use.

**Strengths:**

The paper tackles the challenging task of movie dubbing, considering multiple factors like temporal alignment, emotional expression, speaker timbre preservation, and background noise.
The proposed two-stage method effectively separates the learning of pronunciation from the adaptation to specific movie dubbing contexts. This allows the model to first acquire general pronunciation knowledge before fine-tuning for the nuances of movie dubbing.
The introduction of dedicated modules for prosody consistency learning and duration consistency reasoning addresses crucial aspects of natural and synchronized dubbing, contributing to the model's effectiveness.
The paper reports superior performance compared to state-of-the-art methods on benchmark datasets, demonstrating the practical value of the proposed approach.
The commitment to releasing source code and model checkpoints promotes reproducibility and encourages further research in the field.

**Limitations:**

Generalization to Unseen Scenarios: While the paper demonstrates promising results on benchmark datasets, its generalizability to unseen movie genres, languages, or emotional expressions remains unclear. Further evaluation on diverse and challenging scenarios would strengthen the paper's claims.

Real-World Evaluation: The evaluation focuses on objective metrics and benchmark datasets. Investigating the model's performance in real-world dubbing scenarios and gathering subjective feedback from human evaluators would provide a more comprehensive assessment of its practical utility.

**Suitability:**

3

---

### Official Review · Reviewer_UPfF · 2024-05-24

**Rating:** 5
**Confidence:** 3

**Summary:**

This paper offers an interesting research direction in the domain of movie dubbing, with a commendable level of technical detail and a well-thought-out experimental framework. Further refinement in the areas of originality, methodological depth, experimental validation, and social implications will enhance the manuscript's contribution to the field.

**Strengths:**

Innovation: The paper introduces a novel two-stage framework for movie dubbing that emphasizes prosody and duration consistency, which is a unique approach in the domain of audiovisual synthesis.
Technical Depth: The authors have provided a detailed account of the multi-task pre-training and the prosody and duration consistency learning modules, demonstrating a deep understanding of the challenges in dubbing tasks.
Experimental Design: The paper presents a comprehensive set of experiments on two primary benchmarks, comparing the proposed method with state-of-the-art techniques, which is commendable.

**Limitations:**

1).The Prosody Consistency Learning (PCL) module's use of character emotional features to model dubbing prosody is intriguing. The authors should elaborate on how emotional features are mapped to prosodic attributes and how this process accommodates a range of emotional states.
2). There are some typos in the article.

**Suitability:**

3

---

### Official Review · Reviewer_c2yu · 2024-05-25

**Rating:** 3
**Confidence:** 2

**Summary:**

This paper proposes a two-staged TTS framework to generate speech for silence videos. In the first stage, the phoneme encoder is pretrained on a large corpus with timber-adaptive layer normalization (TALN) and masked language modeling (MLM). In the second stage, facial features are used to predict the pitch and energy, then the duration predictor is replaced by one that considers lip motions.

**Strengths:**

1. This paper has moderate novelty and fits the theme of MM.
2. The presentation of most parts of the paper is good.
3. This paper has sufficient experiments.

**Limitations:**

Overall:
1. Although the experiments are sufficient, some of the results need more explanations and don't quite fit the common sense knowledge.
2. The baselines may not be strong enough. It seems that the baselines in the paper are all trained 'one-stagedly', which means they are directly trained on a small dubbing dataset so they can be easily outperformed. But these baselines can also be trained 'two-stagedly', which means also including a pretraining process on the large dataset. I believe these 'two-stagedly' trained baselines are actually the state-of-the-art approaches for movie dubbing in pactice. Please consider adding the comparison with these approaches to show the real improvement for the movie dubbing industry.

Detailed:
1. About the emotion encoder, attention mechanism is used to capture the emotion features in the video of each phoneme. However, it seems that all frames of the video are used as the keys and values for the  attention mechanism. In the following proposed duration consistency reasoning module, you have already found the matches between the frames and phonemes. So why not use the matched frames as the keys and values for each phoneme in attention mechanism? Or simpler, use max pooling among the matched frames for each phoneme. Please consider adding an ablation study about the structure of emotion encoder.
2. Please add a short description of each baseline used in the experiment in related work.
3. In Table 1 and 2, please consider directly using 'Using the ground truth audio as reference' and 'Using another audio as reference' instead of 'Dub 1.0' and 'Dub 2.0'.
4. In Table 1 and 4, how can WER reach beyond 100% (even 200%)?
5. In Table 1, 2 and 4, the speaker similarities are very low for the many baselines. Please elaborate more on the training setup for the baselines and why the speaker similarities are very low.
6. In Table 2, why are the results of FastSpeech 2 identical under 1.0 and 2.0?
7. The font sizes in Table 3, 4 and 5 are too small.
8. Table 5, 'DCR v.s. Duration Prediction' is not quite suitable for a method name.

**Suitability:**

3

---

### Meta-Review · Area_Chair_kk7S · 2024-06-30

**Recommendation:** Accept (Oral)
**Confidence:** 4

**Metareview:**

This paper proposes a two-stage TTS framework to "dub" (generate speech) for silent videos. In the first stage, the phoneme encoder is pretrained on a large corpus with timbre-adaptive layer normalization (TALN) and masked language modeling (MLM). In the second stage, facial features are used to predict the pitch and energy, then the duration predictor is replaced by one that considers lip motions. Reviewers find the task to be novel, a good to ACM MM and after some clarification questions (answered in the rebuttal) recommend acceptance of the paper.

Reasons to accept:
- This paper has novelty and fits the theme of MM
- The presentation of the paper is clear overall, and reviewers tended to increase their scores in response to the rebuttal
- Conclusions are supported by reasonable ablations and experiments

Reasons to reject:
- The paper is quite dense and authors should further declutter results and focus the presentation.
- The baselines may not be strong enough and even though the proposed method outperforms the tested two-stage baselines (I assume these results will make it to the camera ready), the benefit is less strong. As such, the impact is a bit reduced.